# Impact of the COVID-19 pandemic on skin cancer diagnosis: A population-based study

**Yuka Asai[1], Paul Nguyen[2], Timothy P. Hanna**(ID)[3,4,5]*

**1** Division of Dermatology, Department of Medicine, Queen's University, Kingston, ON, Canada, **2** ICES at Queen's University, Kingston, ON, Canada, **3** Division of Cancer Care and Epidemiology, Cancer Research Institute at Queen's University, Kingston, ON, Canada, **4** Department of Oncology, Queen's University, Kingston, ON, Canada, **5** Department of Public Health Sciences, Queen's University, Kingston, ON, Canada

* tim.hanna@kingstonhsc.ca

## Abstract

### Background

The COVID-19 pandemic has been unprecedented and has led to drastic reductions in non-urgent medical visits. Deferral of these visits may have critical health impact, including delayed diagnosis for melanoma and other skin cancers. We examined the influence of the pandemic on skin biopsy rates in a large population-based cohort.

### Methods

Using a universal health care claims dataset from Ontario, we examined skin biopsies from January 6, 2020 to September 27, 2020, and compared these to the same period for 2019. Those diagnosed with anogenital cancers, younger than 20 years, residing out-of-province and with lapses in coverage were excluded. The sensitivity and specificity of claims diagnoses compared to a validated algorithm to identify keratinocyte carcinoma (KC), or to the cancer registry for melanoma was evaluated. Factors associated with biopsy during the early pandemic were investigated with modified Poisson regression.

### Results

A precipitous drop in total skin biopsies (15% of expected), biopsies for KC (18%) and melanoma (27%) was seen with the onset of COVID-19 cases ($p<0.01$). Claims diagnoses were of high specificity for KC (99%), and for melanoma (98%), though sensitivity was less (61%, 28% respectively). In adjusted analysis, the elderly (80+ years), females and residents of certain regions were less likely to be biopsied during the pandemic. Subsequently, there were substantial improvements in biopsy rates over 10 weeks. However, compared to 2019, a large backlog of expected cases still remained 28 weeks after lockdown (45,710 all biopsy, 9,104 KC, 595 melanoma).

### Interpretation

A drastic reduction in skin biopsies is noted early in the COVID-19 pandemic; this disproportionately affected the elderly, females and certain geographic regions. Though biopsies

**Data Availability Statement:** The minimal dataset is within the paper and Supporting Information files. The full dataset from this study is held securely in coded form at ICES. While data sharing agreements prohibit ICES from making the dataset

publicly available, access may be granted to those who meet pre-specified criteria for confidential access, available at www.ices.on.ca/DAS. The full dataset creation plan and underlying analytic code are available from the authors upon request, understanding that the computer programs may rely upon coding templates or macros that are unique to ICES and therefore may require modification.

**Funding:** TPH holds a research chair provided by the Ontario Institute for Cancer Research through funding provided by the Government of Ontario (#IA-035). This study was supported by ICES, which is funded by an annual grant from the Ontario Ministry of Health and Long-Term Care (MOHLTC). The funders had no role in study design, data collection and analysis, decision to publish, or preparation of the manuscript.

**Competing interests:** According to ICMJE criteria, TPH declares unrestricted research funding for an unrelated project on precision medicine from Roche. YA reports unrelated research grants from the Canadian Institutes of Health Research, the Canadian Dermatology Foundation, AllerGen NCE, Sanofi Canada, Pfizer, Abbvie, and Novartis and honoraria from Janssen, Leo Pharma, Abbvie, Sanofi, Eli Lilly, Pfizer, and Novartis. YA is a clinical trial investigator for Novartis and Leo. This does not alter our adherence to PLOS ON policies on sharing data and materials.

subsequently increased, a large backlog of cases remained after almost half a year. This will have implications for downstream care of skin cancer. Efforts should be made to limit diagnostic delay.

## Introduction

Skin cancer incidence has been increasing in North America over the last decade, mirrored by the number of biopsies for skin cancer [1]. The COVID-19 pandemic has been unprecedented and has led to drastic reductions in non-urgent medical visits. Deferral of these visits may have critical health impact. The impact of the pandemic on skin cancer care has not been well described. There are no reports from Canada, and only two known reports from other countries describing the possible impact. A national report from the Netherlands suggested a drop in skin cancer diagnoses more than for other cancers, though results were not reported according to subtype (e.g. melanoma or squamous cell carcinoma) and data was incomplete [2]. One institutional series from Italy suggested diagnostic delay for melanoma, though generalizability is limited given the narrow population coverage [3]. We hypothesized that the impact on skin cancer care in a Canadian population-level sample would be substantial, with potential differences between skin cancer types and some patient groups. Therefore, we examined the influence of the COVID-19 pandemic on the diagnosis of skin cancer via biopsy in a large population-based cohort in a Canadian context, stratifying according melanoma and non-melanoma cancer types.

## Methods

### Study design

This was a retrospective population-based cohort study.

### Study population

Using a universal health care claims dataset from Canada's most populous province, Ontario (N = 14.7 million), we undertook a population-based study examining skin biopsies from January 6, 2020 to September 27, 2020, and compared these to the same period for 2019. Our primary analysis focused on the early lockdown, and covered January 6, 2020 to April 19, 2020. Those diagnosed with anogenital cancers, younger than 20 years, residing out-of-province, and those who had lapses with provincial medical coverage were excluded (Fig 1, S1 Appendix). This study followed the Strengthening the Reporting of Observational Studies in Epidemiology (STROBE) reporting guideline for cohort studies.

### Data sources

Claims data from the Ontario Health Insurance Plan (OHIP) were utilized for outcome measurement. OHIP records are updated monthly with claims validated and processed by the Ontario Ministry of Health and Long-Term Care (MOHLTC). As described below, data from the Ontario Cancer Registry (OCR) were used to evaluate the specificity of melanoma claims to melanoma diagnoses. The OCR is known for its high level of population coverage [4], with annual capture rates generally 94% or more for melanoma in a major validation study [5]. Age and sex were determined from the Registered Persons Database maintained by MOHLTC. Area-level income quintiles were determined based on quintile rankings of neighbourhood

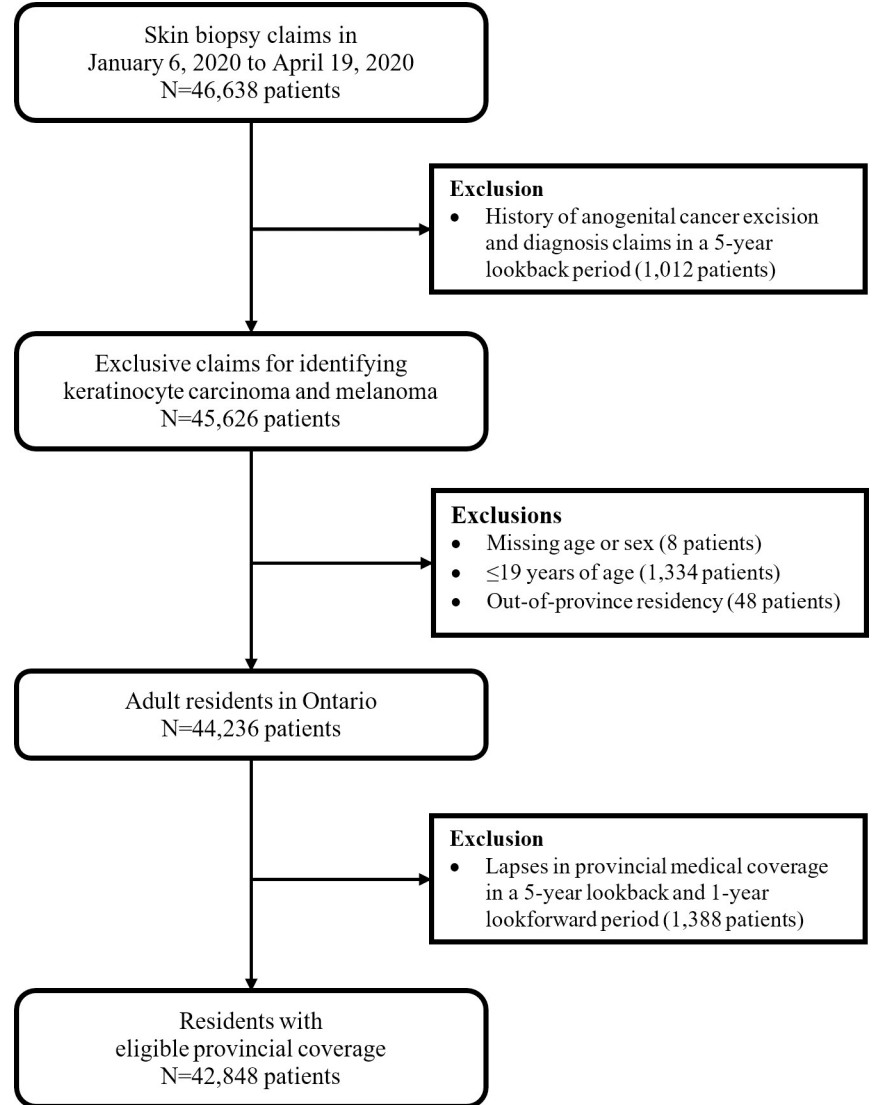

**Fig 1. Flow diagram for selection of study cohort of skin biopsy claims from January 6, 2020 to April 19, 2020.**

average income within each census metropolitan area or census agglomeration. Comorbidity was based on the Elixhauser comorbidity index using CIHI Discharge Abstract Database (DAD) and Same Day Surgery (SDS) data available up to March 31, 2020, with zero indicating no identified comorbidity and one point for each additional comorbidity in the index [6]. A five-year lookback window for admissions was utilized. Diagnostic codes for cancer metastasis or solid tumor without metastasis were not included in the comorbidity measure. Rurality was based on the 2008 Rurality Index for Ontario scale, ranging from 0 to 100, with 100 indicating the highest degree of rurality [7]. The biopsying physician main specialty was based on the ICES Physician Database.

## Outcomes

Claims for first skin biopsy extracted on July 21, 2020 in each study interval (January 7 to April 21, 2019 and January 6 to April 19, 2020) for each of the following were considered: 1) any

diagnosis (2019: N = 57,744; 2020: N = 42,848), 2) keratinocyte carcinoma (KC) [basal cell carcinoma (BCC) or squamous cell carcinoma (SCC); ICD-9: 173)] (2019: N = 13,734; 2020: N = 10,776) and 3) melanoma (ICD-9: 172) (2019: N = 910; 2020: N = 736). Claims diagnoses were investigated as they were available in a more timely fashion compared to OCR, and provided more complete information on non-melanoma skin cancers. Trends in biopsies beyond the early lockdown period (weeks 11 to 15 of 2020) were quantified up to week 38 of 2020 based on administrative data extracted on February 11, 2021.

## Statistical analysis

Statistical analyses were conducted using the SAS software version 9.4 (SAS Institute Inc., Cary, NC). Two-sided statistical significance was a p-value<0.05. Descriptive statistics of the patient socio-demographics and general health characteristics were first generated for the study cohort of skin biopsy claims for the first 15 weeks (starting on Monday) of 2020, and for comparison, the first 15 weeks of 2019. Results were stratified by the pre-COVID-19 (Weeks 1–10) and COVID-19 (Weeks 11–15) periods defined by the Declaration of Emergency in Ontario enacted on March 17, 2020.

For comparing proportional change in biopsy rates between KC and melanoma outcomes in the COVID-19 era, Pearson's chi-squared test was used to evaluate the difference between 2019 and 2020. To evaluate sensitivity and specificity of a KC claims diagnosis to a confirmed cancer diagnosis, we compared OHIP biopsy claims for July 2018-June 2019 to an algorithm validated in Ontario using pathology reports [8]. This algorithm was not directly used as an outcome for our study, as we lacked the required 6-month follow-up period. For melanoma, we compared claims diagnoses to the provincial cancer registry [5]. The period of selection for this analysis was based on the availability of complete registry data. Modified Poisson regression was then used to evaluate the patient characteristics between the COVID-19 and pre-COVID-19 eras for the first 15 weeks of 2020. The potential predictors that were modeled in the unadjusted analyses were included in the full adjusted model.

## Ethics approval

This study was approved and overseen by the Queen's University Health Sciences Research Ethics Board (Study ONGY-579-20), with a waiver of informed consent based on the Personal Health Information Protection Act, Section 44(1), which allows waivers of informed consent for research meeting criteria including whether the research can be accomplished without the requested personal health information, research that is in the public interest, safeguards for privacy, and impracticality of obtaining informed consent. Analyses were conducted using anonymized encoded data sources. Administrative records for the study population covered January 6, 2020 to September 27, 2020, and the same period for 2019, with a 5-year look back to measure the comorbidity index, while records for sensitivity and specificity evaluation covered July 1, 2018 to June 30, 2019.

## Results

There were 42,848 with a first skin biopsy in weeks 1 to 15 of 2020 (Fig 1). In this cohort, the median age was 63 years (interquartile range (IQR): 50–74 years), and 51% were female. This was the same for the 57,744 people with a first skin biopsy in weeks 1 to 15 of 2019 (S1 Appendix). Many patient factors (e.g., age, income quintiles, rurality, Elixhauser comorbidity index) were similar between the pre-COVID-19 and COVID-19 periods in 2020 (Table 1).

A precipitous drop in the number of skin biopsies was observed starting in mid-March 2020, coinciding with the report of COVID-19 in Ontario and announcement of emergency

**Table 1. Patient characteristics for all biopsy claims for the first 15 weeks (starting on Monday) of 2019 and 2020.**

| Patient Characteristics | 2019 | | | 2020 | | |
|---|---|---|---|---|---|---|
| | Weeks 1–10 N = 37,938 | Weeks 11–15 N = 19,806 | Total N = 57,744 | Pre-COVID-19 Weeks 1–10 N = 39,224 | COVID-19 Weeks 11–15 N = 3,624 | Total N = 42,848 |
| Age | | | | | | |
| Mean ± SD | 61.17 ± 16.94 | 61.42 ± 17.08 | 61.26 ± 16.99 | 61.73 ± 16.90 | 60.82 ± 16.52 | 61.65 ± 16.87 |
| Median (IQR) | 63 (50–74) | 63 (50–74) | 63 (50–74) | 63 (50–74) | 63 (50–73) | 63 (50–74) |
| Age (categorized) | | | | | | |
| 20–59 | 16,353 (43.10%) | 8,428 (42.55%) | 24,781 (42.92%) | 16,370 (41.73%) | 1,536 (42.38%) | 17,906 (41.79%) |
| 60–69 | 8,495 (22.39%) | 4,361 (22.02%) | 12,856 (22.26%) | 8,853 (22.57%) | 887 (24.48%) | 9,740 (22.73%) |
| 70–79 | 7,519 (19.82%) | 4,049 (20.44%) | 11,568 (20.03%) | 7,961 (20.30%) | 742 (20.47%) | 8,703 (20.31%) |
| 80+ | 5,571 (14.68%) | 2,968 (14.99%) | 8,539 (14.79%) | 6,040 (15.40%) | 459 (12.67%) | 6,499 (15.17%) |
| Sex | | | | | | |
| Female | 19,484 (51.36%) | 10,244 (51.72%) | 29,728 (51.48%) | 20,138 (51.34%) | 1,725 (47.60%) | 21,863 (51.02%) |
| Male | 18,454 (48.64%) | 9,562 (48.28%) | 28,016 (48.52%) | 19,086 (48.66%) | 1,899 (52.40%) | 20,985 (48.98%) |
| Income quintiles[1] | | | | | | |
| 1 | 5,674 (14.96%) | 2,901 (14.65%) | 8,575 (14.85%) | 5,625 (14.34%) | 521 (14.38%) | 6,146 (14.34%) |
| 2 | 6,891 (18.16%) | 3,559 (17.97%) | 10,450 (18.10%) | 6,906 (17.61%) | 626 (17.27%) | 7,532 (17.58%) |
| 3 | 7,405 (19.52%) | 3,860 (19.49%) | 11,265 (19.51%) | 7,674 (19.56%) | 730 (20.14%) | 8,404 (19.61%) |
| 4 | 7,934 (20.91%) | 4,206 (21.24%) | 12,140 (21.02%) | 8,515 (21.71%) | 803 (22.16%) | 9,318 (21.75%) |
| 5 | 9,978 (26.30%) | 5,255 (26.53%) | 15,233 (26.38%) | 10,447 (26.63%) | 935 (25.80%) | 11,382 (26.56%) |
| Rurality Index for Ontario[1] | | | | | | |
| Urban (0–9) | 25,017 (65.94%) | 12,816 (64.71%) | 37,833 (65.52%) | 25,486 (64.98%) | 2,244 (61.92%) | 27,730 (64.72%) |
| Suburban (10–39) | 9,152 (24.12%) | 4,910 (24.79%) | 14,062 (24.35%) | 9,866 (25.15%) | 982 (27.10%) | 10,848 (25.32%) |
| Rural (40+) | 3,528 (9.30%) | 1,943 (9.81%) | 5,471 (9.47%) | 3,599 (9.18%) | 360 (9.93%) | 3,959 (9.24%) |
| Place of residence (LHIN) | | | | | | |
| 01 | 2,103 (5.54%) | 1,093 (5.52%) | 3,196 (5.53%) | 2,351 (5.99%) | 155 (4.28%) | 2,506 (5.85%) |
| 02 | 4,075 (10.74%) | 2,098 (10.59%) | 6,173 (10.69%) | 4,173 (10.64%) | 323 (8.91%) | 4,496 (10.49%) |
| 03 | 2,156 (5.68%) | 1,073 (5.42%) | 3,229 (5.59%) | 2,213 (5.64%) | 184 (5.08%) | 2,397 (5.59%) |
| 04 | 4,592 (12.10%) | 2,440 (12.32%) | 7,032 (12.18%) | 4,296 (10.95%) | 453 (12.50%) | 4,749 (11.08%) |
| 05 | 1,403 (3.70%) | 718 (3.63%) | 2,121 (3.67%) | 1,495 (3.81%) | 99 (2.73%) | 1,594 (3.72%) |
| 06 | 2,931 (7.73%) | 1,483 (7.49%) | 4,414 (7.64%) | 3,028 (7.72%) | 308 (8.50%) | 3,336 (7.79%) |
| 07 | 3,286 (8.66%) | 1,740 (8.79%) | 5,026 (8.70%) | 3,552 (9.06%) | 270 (7.45%) | 3,822 (8.92%) |
| 08 | 3,997 (10.54%) | 2,082 (10.51%) | 6,079 (10.53%) | 4,239 (10.81%) | 331 (9.13%) | 4,570 (10.67%) |
| 09 | 3,462 (9.13%) | 1,813 (9.15%) | 5,275 (9.14%) | 3,652 (9.31%) | 300 (8.28%) | 3,952 (9.22%) |
| 10 | 1,789 (4.72%) | 891 (4.50%) | 2,680 (4.64%) | 1,751 (4.46%) | 186 (5.13%) | 1,937 (4.52%) |
| 11 | 3,764 (9.92%) | 2,069 (10.45%) | 5,833 (10.10%) | 3,957 (10.09%) | 358 (9.88%) | 4,315 (10.07%) |
| 12 | 2,036 (5.37%) | 1,081 (5.46%) | 3,117 (5.40%) | 2,215 (5.65%) | 388 (10.71%) | 2,603 (6.07%) |
| 13 | 1,826 (4.81%) | 947 (4.78%) | 2,773 (4.80%) | 1,824 (4.65%) | 214 (5.91%) | 2,038 (4.76%) |
| 14 | 518 (1.37%) | 278 (1.40%) | 796 (1.38%) | 478 (1.22%) | 55 (1.52%) | 533 (1.24%) |

(*Continued*)

**Table 1.** (Continued)

| Patient Characteristics | 2019 | | | 2020 | | |
|---|---|---|---|---|---|---|
| | Weeks 1–10 N = 37,938 | Weeks 11–15 N = 19,806 | Total N = 57,744 | Pre-COVID-19 Weeks 1–10 N = 39,224 | COVID-19 Weeks 11–15 N = 3,624 | Total N = 42,848 |
| Elixhauser comorbidity index[2] | | | | | | |
| 0 | 30,176 (79.54%) | 15,875 (80.15%) | 46,051 (79.75%) | 31,265 (79.71%) | 2,944 (81.24%) | 34,209 (79.84%) |
| 1–2 | 5,617 (14.81%) | 2,824 (14.26%) | 8,441 (14.62%) | 5,755 (14.67%) | 491 (13.55%) | 6,246 (14.58%) |
| 3+ | 2,145 (5.65%) | 1,107 (5.59%) | 3,252 (5.63%) | 2,204 (5.62%) | 189 (5.22%) | 2,393 (5.58%) |
| Physician specialty billing biopsy claims | | | | | | |
| Dermatology | 19,061 (50.24%) | 9,908 (50.03%) | 28,969 (50.17%) | 19,982 (50.94%) | 1,810 (49.94%) | 21,792 (50.86%) |
| GP/FP | 10,963 (28.90%) | 6,026 (30.43%) | 16,989 (29.42%) | 11,375 (29.00%) | 1,058 (29.19%) | 12,433 (29.02%) |
| General surgery | 2,287 (6.03%) | 1,247 (6.30%) | 3,534 (6.12%) | 2,354 (6.00%) | 185 (5.10%) | 2,539 (5.93%) |
| Plastic surgery | 2,775 (7.31%) | 1,327 (6.70%) | 4,102 (7.10%) | 2,560 (6.53%) | 301 (8.31%) | 2,861 (6.68%) |
| Otolaryngology | 1,369 (3.61%) | 601 (3.03%) | 1,970 (3.41%) | 1,273 (3.25%) | 122 (3.37%) | 1,395 (3.26%) |
| Other | 1,483 (3.91%) | 697 (3.52%) | 2,180 (3.78%) | 1,680 (4.28%) | 148 (4.08%) | 1,828 (4.27%) |

**Abbreviations:**

SD: standard deviation, IQR: interquartile range, LHIN: Local Health Integration Network, GP/FP: general practitioner/family practitioner.

**Notes:**

[1]Column percentages may not sum to 100% due to missing data.

[2]Diagnostic codes for cancer metastasis or solid tumor without metastasis were excluded from the comorbidity score.

measures. Compared to 2019, only 15% of the expected numbers of all skin biopsies, and those undertaken in conjunction with a diagnostic code for KC (18%) and melanoma (27%) were observed (Fig 2). The drop in KC-associated biopsies was greater than for melanoma ($p<0.01$).

We found our results of high specificity compared to a validated algorithm [8] to identify KC (87% accuracy, 61% sensitivity, 99% specificity). Similarly, ICD-9 claim codes are highly specific for melanoma, compared to the OCR (98% accuracy, 28% sensitivity, 98% specificity). Importantly, although claims-based diagnoses lacked pathological confirmation, regardless of claim diagnosis, biopsies dropped dramatically.

We investigated whether any group of patients were disproportionately affected during the early lockdown (weeks 11 to 15 of 2020). In the adjusted analysis, elderly patients (80+ years vs. 20–59 years, relative risk (RR) with 95% confidence interval (CI): 0.81 (0.73–0.90), $p<0.01$) and patients residing in Southwestern Ontario (01, 02) and the Greater Toronto Area (05, 07, 08, 09) compared to the South East LHIN (10) were less likely to have a skin biopsy in the COVID-19 era (Table 2). Furthermore, male patients (RR: 1.16 (1.09–1.23), $p<0.01$) and patients biopsied by plastic surgeons (plastic surgeon vs. other specialists, RR: 1.37 (1.13–1.66), $p<0.01$) were more likely to have a skin biopsy in the COVID-19 era.

Keratinocyte cancer-associated biopsies showed similar results between the COVID-19 and pre-COVID-19 periods: Elderly patients (70–79 years vs. 20–59 years, RR: 0.83 (0.70–0.99), $p = 0.03$; 80+ years vs. 20–59 years, RR: 0.69 (0.57–0.83), $p<0.01$) and patients residing in Southwestern Ontario (02) and the Greater Toronto Area (05, 08, 09) compared to the South East LHIN (10) were less likely to have a KC claims diagnosis in the COVID-19 era, while male patients (RR: 1.25 (1.11–1.41), $p<0.01$) were more likely to have KC-associated biopsies in the COVID-19 era compared with pre-COVID-19 era (S2 and S3 Appendices).

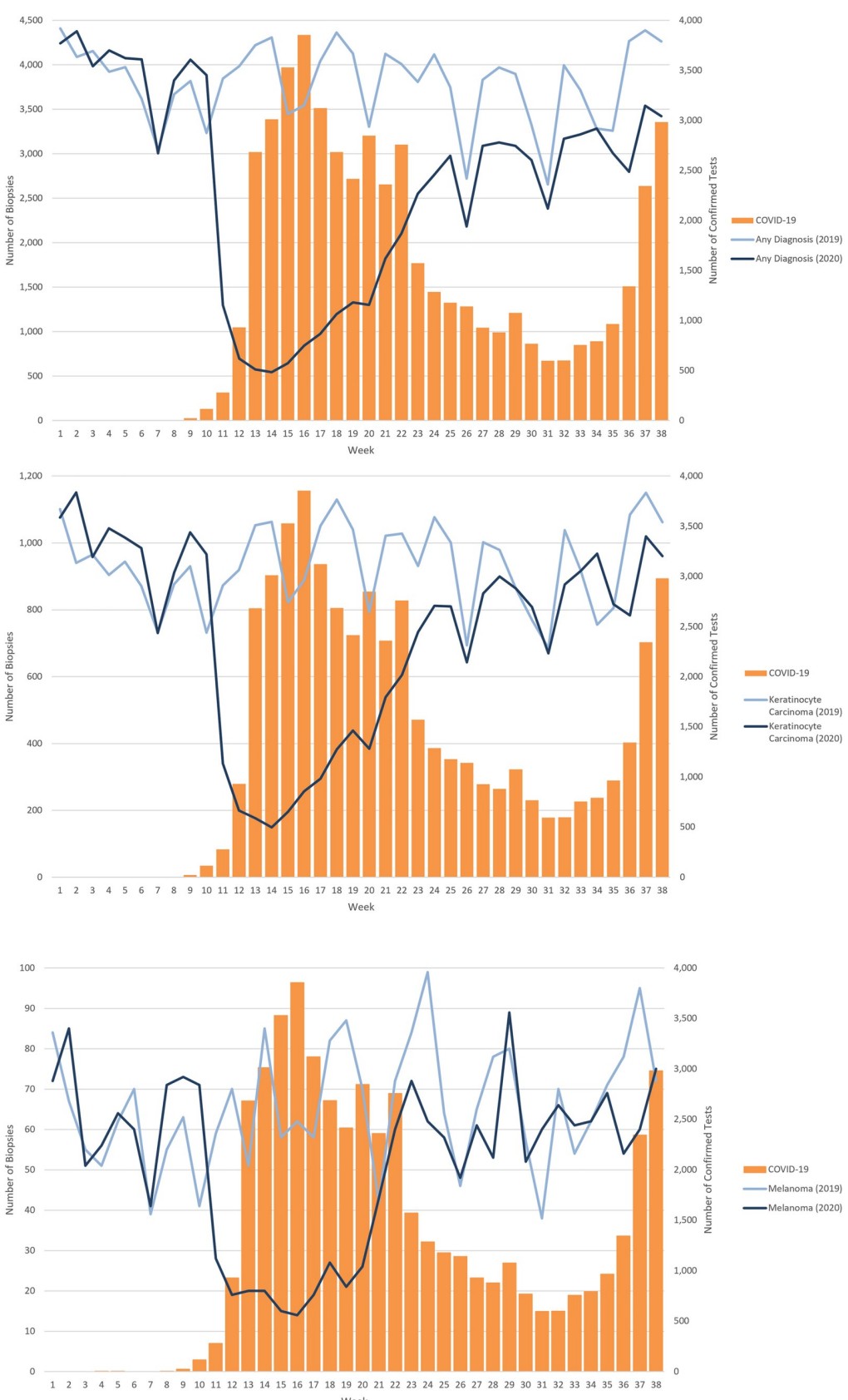

**Fig 2. A.** Time trends of all skin biopsy claims for the first 38 weeks (starting on Monday) of 2019 and 2020 in Ontario in relation to COVID-19 cases in Ontario, Canada. **B.** Time trends of skin biopsy claims associated with a diagnosis of keratinocyte carcinoma for the first 38 weeks (starting on Monday) of 2019 and 2020 in relation to COVID-19 cases in Ontario, Canada. **C.** Time trends of skin biopsy claims associated with a diagnosis of melanoma for the first 38 weeks (starting on Monday) of 2019 and 2020 in Ontario in relation to COVID-19 cases in Ontario, Canada.

For melanoma-associated biopsies, there were similar trends for age (80+ years vs. 20–59 years, RR: 0.70 (0.36–1.36), $p$ = 0.29) and male sex (RR: 1.15 (0.77–1.72), $p$ = 0.49), though findings were not significant (S4 and S5 Appendices). Patients residing in higher income neighborhoods (5 vs. 1, RR: 2.59 (1.20–5.60), $p$ = 0.02) and with higher comorbidities (3+ vs. 0, RR: 2.01 (1.09–3.71), $p$ = 0.03) were more likely to have a biopsy in the COVID-19 era while patients residing in suburban areas were less likely (suburban vs. rural, RR: 0.54 (0.30–0.98), $p$ = 0.04). Biopsies for melanoma in the COVID-19 era were proportionately less likely to be performed by dermatologists (dermatologist vs. otolaryngologist or other specialists, RR: 0.52 (0.30–0.92), $p$ = 0.02) or GPs/FPs (GP/FP vs. otolaryngologist or other specialists, RR: 0.36 (0.17–0.73), $p$ = 0.01) compared with the pre-COVID-19 period.

Following the precipitous drops in skin biopsies seen in the early lockdown (weeks 11–15), a slow increase in numbers of skin biopsies was observed over the following 10 weeks. However, following lockdown, skin biopsy rates in most weeks of 2020 did not meet those of the same week in 2019 (Fig 2A–2C). As skin cancer incidence is expected to increase from year to year, this deficit is noticeable, with 45,710 fewer cases for all skin biopsies, 9,104 for KC and 595 for melanoma between weeks 11–38 of 2020, compared to 2019 weeks 11–38.

## Interpretation

Using a claims-based method in a Canadian population, large drops in observed rates were seen with the onset of COVID-19 cases in Ontario for the total number of skin biopsies (15% of expected), and skin biopsies specifically performed in conjunction with a diagnostic code for keratinocyte carcinoma (18% of expected) and melanoma (27% of expected) ($p$<0.01). A differential change between KC biopsies and melanoma biopsies suggests a triage effect, though changes in skin biopsy in all categories was large, and precipitous. As overall demographic characteristics of the 2020 and 2019 data for the same period were similar, it is evident some patient groups were disproportionately affected during the lockdown, with biopsies less likely among females, the elderly (80+ years), and residents of certain regions in the weeks after the shutdown than in the prior 10 weeks. Moreover, despite increases in biopsies following the early lockdown, a large backlog in expected skin biopsies persisted to the end of the study period, which was almost half a year beyond the early lockdown (45,710 all biopsy, 9,104 KC, 595 melanoma).

Our findings provide direct evidence of the impact of COVID-19 in skin cancer care in a Canadian setting, and augment the very limited data on impact of the pandemic in this setting. A report from the Netherlands suggests our findings may be part of a broader trend, with observed drops reported in diagnosis of non-basal cell carcinoma skin cancers, in a limited data set [2]. A triage effect is suggested by our more detailed population data; there was a significant difference in proportional drop in biopsy, which was greater for biopsies associated with KC diagnoses compared with melanoma. A single-institution report from Italy suggests that diagnostic delay in melanoma during the pandemic may be translating into greater thickness of melanoma at diagnosis, though this requires broader confirmation across jurisdictions [3].

Biopsy trends observed in our population-based sample may be explained by a number of factors, such as: 1) decreased early detection due to public health regulations limiting non-

**Table 2. Relative risks for comparing all skin biopsy claims between COVID-19 (Weeks 11–15) and pre-COVID-19 (Weeks 1–10) periods for the first 15 weeks (starting on Monday) of 2020.**

| Patient Characteristics | Unadjusted Analysis | | Adjusted Analysis[1] | |
|---|---|---|---|---|
| | RR (95% CI) | P-Value | RR (95% CI) | P-Value |
| Age (categorized) | | | | |
| 20–59 | Reference | | Reference | |
| 60–69 | 1.06 (0.98–1.15) | 0.147 | 1.03 (0.95–1.11) | 0.496 |
| 70–79 | 0.99 (0.91–1.08) | 0.886 | 0.96 (0.88–1.05) | 0.381 |
| 80+ | 0.83 (0.75–0.91) | <0.001 | 0.81 (0.73–0.90) | <0.001 |
| Sex | | | | |
| Female | Reference | | Reference | |
| Male | 1.15 (1.08–1.22) | <0.001 | 1.16 (1.09–1.23) | <0.001 |
| Income quintiles | | | | |
| 1 | Reference | | Reference | |
| 2 | 0.99 (0.88–1.11) | 0.850 | 0.99 (0.89–1.11) | 0.881 |
| 3 | 1.03 (0.92–1.15) | 0.582 | 1.02 (0.91–1.13) | 0.751 |
| 4 | 1.03 (0.92–1.14) | 0.649 | 1.00 (0.90–1.11) | 0.947 |
| 5 | 0.98 (0.88–1.08) | 0.648 | 0.95 (0.85–1.05) | 0.294 |
| Rurality Index for Ontario | | | | |
| Urban (0–9) | 0.89 (0.80–0.99) | 0.031 | 1.01 (0.90–1.14) | 0.822 |
| Suburban (10–39) | 1.00 (0.89–1.12) | 0.939 | 1.11 (0.98–1.25) | 0.105 |
| Rural (40+) | Reference | | Reference | |
| Place of residence (LHIN) | | | | |
| 01 | 0.65 (0.53–0.80) | <0.001 | 0.65 (0.53–0.80) | <0.001 |
| 02 | 0.75 (0.63–0.89) | 0.001 | 0.80 (0.67–0.95) | 0.012 |
| 03 | 0.80 (0.66–0.98) | 0.028 | 0.84 (0.69–1.02) | 0.075 |
| 04 | 1.00 (0.85–1.18) | 0.982 | 1.03 (0.87–1.21) | 0.723 |
| 05 | 0.65 (0.51–0.82) | <0.001 | 0.67 (0.52–0.85) | 0.001 |
| 06 | 0.97 (0.81–1.15) | 0.714 | 1.04 (0.86–1.24) | 0.709 |
| 07 | 0.74 (0.62–0.89) | 0.001 | 0.78 (0.65–0.94) | 0.010 |
| 08 | 0.76 (0.64–0.91) | 0.002 | 0.79 (0.66–0.94) | 0.009 |
| 09 | 0.80 (0.67–0.95) | 0.011 | 0.81 (0.68–0.97) | 0.022 |
| 10 | Reference | | Reference | |
| 11 | 0.87 (0.74–1.04) | 0.118 | 0.93 (0.78–1.11) | 0.414 |
| 12 | 1.56 (1.33–1.85) | <0.001 | 1.67 (1.41–1.97) | <0.001 |
| 13 | 1.08 (0.89–1.31) | 0.440 | 1.15 (0.95–1.39) | 0.157 |
| 14 | 1.14 (0.85–1.53) | 0.381 | 1.21 (0.90–1.63) | 0.212 |
| Elixhauser comorbidity index[2] | | | | |
| 0 | Reference | | Reference | |
| 1–2 | 0.92 (0.84–1.01) | 0.079 | 0.93 (0.84–1.02) | 0.123 |
| 3+ | 0.92 (0.79–1.06) | 0.228 | 0.96 (0.83–1.11) | 0.561 |
| Physician specialty billing biopsy claims | | | | |
| Dermatology | 1.02 (0.87–1.20) | 0.789 | 1.06 (0.90–1.24) | 0.510 |
| GP/FP | 1.04 (0.88–1.23) | 0.630 | 1.02 (0.86–1.20) | 0.833 |
| General surgery | 0.89 (0.72–1.09) | 0.261 | 0.81 (0.66–1.00) | 0.054 |
| Plastic surgery | 1.29 (1.07–1.56) | 0.007 | 1.37 (1.13–1.66) | 0.001 |
| Otolaryngology | 1.09 (0.86–1.37) | 0.478 | 1.05 (0.83–1.32) | 0.688 |

(*Continued*)

**Table 2.** (Continued)

| Patient Characteristics | Unadjusted Analysis | | Adjusted Analysis[1] | |
|---|---|---|---|---|
| | RR (95% CI) | P-Value | RR (95% CI) | P-Value |
| Other | Reference | | Reference | |

**Abbreviations:**

RR: relative risk, CI: confidence interval, LHIN: Local Health Integration Network, GP/FP: general practitioner/family practitioner.

**Notes:**

[1]The full adjusted model contains all variables in the unadjusted analyses.

[2]Diagnostic codes for cancer metastasis or solid tumor without metastasis were excluded from the comorbidity score.

urgent health care provision and social distancing, 2) decreased access to primary health care and specialist care, 3) lack of transportation services, 4) childcare or eldercare responsibilities, 5) limitations of telehealth, and 6) patients refusing to seek care or refusing care due to perceived infection risk.

The variable impact of the early lockdown on skin biopsies may be explained by some of these factors. We observed females and the elderly (80+ years) were less likely to be biopsied during the pandemic, as were residents of certain regions in the province. We also observed a higher proportion of biopsies were performed by plastic surgeons during the initial months of the COVID-19 epidemic in Ontario than by general practitioners, dermatologists or other specialists. Limitations in non-urgent health care provision probably contributed to the proportionally greater decline in biopsies by general practitioners and dermatologists. As plastic surgeons may manage more advanced lesions, their biopsy rates may have stayed proportionately higher compared to other types of health practitioners. Reasons for fewer biopsies among female patients are unclear, but may be multifactorial. This may relate in part to differences in caregiver burden between male and female groups in the cohort. Behavioral differences in care seeking between males and females have also been described in other settings [9]. In a population-based study of melanoma in Ontario, we observed that males were more likely to present with advanced stage disease [10]. If males in our cohort had a higher burden of advanced skin cancer, proportionately more cases would have had concerning findings or symptoms, possibly leading to faster diagnostic biopsy.

We observed that for melanoma, the presence of multiple comorbidities and residence in higher income quintile neighbourhoods were associated with a much greater likelihood for biopsy during the early lockdown. The association with comorbidity may relate to diagnoses captured during admissions for management of advanced melanoma (e.g. for biopsy of lymph node metastasis or symptomatic metastatic disease). Some biopsies for melanoma among the high-comorbidity group may have been for incidental findings during admissions for comorbid conditions, though the same pattern was not seen for keratinocyte carcinoma. The association with income quintile may relate to greater availability of services for skin biopsy in higher income quintile regions [11] or greater self-advocacy or melanoma awareness for those living in these regions. Notably, we previously observed a lower risk of advanced melanoma in less deprived regions in Ontario [10].

The decreased likelihood for biopsy in certain LHINs may be related to the incidence rate of COVID-19 in those regions. Both the Greater Toronto Area and the Southwestern region had higher overall rates of cases of COVID-19 compared to the Southeast LHIN, and decreased likelihood of biopsy.

In the absence of a diagnostic biopsy, it is difficult to ascertain the risk posed by a suspicious skin lesion; a lesion thought to be BCC could be melanoma. The impact of pandemic-related delay on skin cancer outcomes is yet unknown but is a cause for concern given the strong

relationship between early detection and skin cancer outcomes. Evidence suggests a possible association of delay in melanoma diagnosis with stage progression and survival [12], and between delay in SCC excision and tumour size, which is known to correlate with metastasis [13]. Moreover, the observed impact on skin cancer diagnosis is expected to be compounded by waiting times required for backlog management in the post-lockdown period, given that our results suggest a large number of cases with deferred biopsies, which will be added to the normal influx of new cases arising on a daily basis. Of great concern, we found that almost half a year after lockdown measures started, there were still large deficits in expected numbers of skin biopsies, with 45,710 fewer cases for all skin biopsies, 9,104 for KC and 595 for melanoma between weeks 11–38 of 2020, compared to 2019 weeks 11–38. This comparison does not take into account the normal expected annual increases in cancer incidence.

Disparities in access to timely diagnosis of melanoma in Ontario are suggested by a recent population based study [10], and Ontario is facing a dermatologist shortage that will likely increase with attrition of the work force [14], compounded by a longstanding deficiency in primary care providers. Canada as a whole and other countries face similar human resource issues [15–17]. Health systems should ensure the timely diagnosis of skin cancer; this should be reflected by public health messages encouraging individuals with suspicious skin lesions to seek evaluation within an appropriate timeline.

There are limitations to our study. We were not able to directly measure changes in pathologic diagnoses of skin cancer or stage as complete data is not available. However, we were able to measure changes in skin biopsies, which ultimately will lead to changes in pathologic cancer diagnoses. Moreover, we found our claims algorithm highly specific compared to a validated algorithm identifying pathologically proven keratinocyte cancer diagnoses, and melanoma diagnoses in the Ontario cancer registry. Our data on comorbidity was limited to an administrative data algorithm without chart data abstraction, and CIHI data defining comorbidity was incomplete, notably for the last two and a half weeks of our study period. Given the use of a five-year lookback for comorbid diagnoses in CIHI, the impact on overall findings is expected to be small. Our results could under-estimate the magnitude of delayed cancer diagnosis as shown by the moderate sensitivity of our approach for KC and low sensitivity for melanoma. Melanoma may be more difficult to correctly identify clinically, especially for generalists. Moreover, some melanoma claims are for recurrences, or *in-situ* melanoma that is not captured in OCR.

## Conclusion

A drastic reduction in number of skin biopsies is noted early in the COVID-19 pandemic; this disproportionately affected females, the elderly and residents of certain regions. A large backlog of skin biopsies was observed during the pandemic, persisting well after initial lockdown measures were instated. This will have implications for downstream care of skin cancer. We recommend that public health messages encourage individuals with suspicious skin lesions to seek evaluation in a timely fashion, and that this is welcome and encouraged. Groups at greater risk of delay should receive special attention. Moreover, health systems should increase the priority of unbiopsied skin lesions, and ensure sufficient resources for management of the expected backlog in cases.

## Supporting information

**S1 Appendix. Flow diagram for selection of study cohort of skin biopsy claims from January 7, 2019 to April 21, 2019.**
(DOCX)

**S2 Appendix. Patient characteristics for skin biopsy claims associated with a diagnosis of keratinocyte carcinoma for the first 15 weeks (starting on Monday) of 2019 and 2020.**
(DOCX)

**S3 Appendix. Relative risks for comparing skin biopsy claims associated with a diagnosis of keratinocyte carcinoma between COVID-19 (Weeks 11–15) and pre-COVID-19 (Weeks 1–10) periods for first 15 weeks (starting on Monday) of 2020.**
(DOCX)

**S4 Appendix. Patient characteristics for skin biopsy claims associated with a diagnosis of melanoma for the first 15 weeks (starting on Monday) of 2019 and 2020.**
(DOCX)

**S5 Appendix. Relative risks for comparing skin biopsy claims associated with a diagnosis of melanoma between COVID-19 (Weeks 11–15) and pre-COVID-19 (Weeks 1–10) periods for first 15 weeks (starting on Monday) of 2020.**
(DOCX)

## Acknowledgments

The opinions, results and conclusions reported in this paper are those of the authors and are independent from the funding sources. No endorsement by ICES or the Ontario Ministry of Health and Long-Term Care (MOHLTC) is intended or should be inferred. Parts of this material are based on data and/or information compiled and provided by CIHI. However, the analyses, conclusions, opinions and statements expressed in the material are those of the author(s), and not necessarily those of CIHI. Parts of this material are based on data and information provided by Cancer Care Ontario (CCO). The opinions, results, view, and conclusions reported in this paper are those of the authors and do not necessarily reflect those of CCO. No endorsement by CCO is intended or should be inferred. These data were linked using unique encoded identifiers and analyzed at ICES.

The Ontario dataset for confirmed positive cases of COVID-19 was obtained from the Ontario Open Catalogue. All materials are licensed under the Open Government Licence–Ontario.

We also acknowledge Chao Xue for her administrative support.

## Author Contributions

**Conceptualization:** Timothy P. Hanna.

**Data curation:** Paul Nguyen, Timothy P. Hanna.

**Formal analysis:** Paul Nguyen.

**Funding acquisition:** Timothy P. Hanna.

**Investigation:** Yuka Asai, Timothy P. Hanna.

**Methodology:** Yuka Asai, Paul Nguyen, Timothy P. Hanna.

**Project administration:** Yuka Asai, Timothy P. Hanna.

**Resources:** Timothy P. Hanna.

**Software:** Paul Nguyen.

**Supervision:** Timothy P. Hanna.

**Validation:** Paul Nguyen, Timothy P. Hanna.

**Visualization:** Timothy P. Hanna.

**Writing – original draft:** Yuka Asai.

**Writing – review & editing:** Yuka Asai, Paul Nguyen, Timothy P. Hanna.

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
