## [Decision Letter · Decision Letter 0]

5 Feb 2021

PONE-D-20-37376

Impact of the COVID-19 pandemic on skin cancer diagnosis: a population-based study

PLOS ONE

Dear Dr. Hanna

Thank you for submitting your manuscript to PLOS ONE. After careful consideration, we feel that it has merit but does not fully meet PLOS ONE’s publication criteria as it currently stands. Therefore, we invite you to submit a revised version of the manuscript that addresses the points raised during the review process.

Even if both reviewers found the manuscript interesting, some additional evaluation and comments on the obtained data would be useful to improve the manuscript's relevance. 

We look forward to receiving your revised manuscript.

Kind regards,

Mauro Picardo, M.D.

Academic Editor

PLOS ONE

Journal Requirements:

2. Please note that PLOS journals require authors to make all data underlying the findings described in their manuscript fully available without restriction at the time of publication. When specific legal or ethical requirements prohibit public sharing of a dataset, authors must indicate how researchers may obtain access to the data. PLOS journals will not consider manuscripts for which the following factors influence ability to share data:

- Authors will not share data because of personal interests, such as patents or potential future publications.

- The conclusions depend solely on the analysis of proprietary data, whether these data are owned by the authors, by their funders or institutions, or by other parties.

Therefore, please update your Data Availability statement to indicate how other researchers may gain access to the underlying data reported in the manuscript.

For more information, please see: https://journals.plos.org/plosone/s/data-availability

3. In your ethics statement in the manuscript and in the online submission form, please ensure that you have discussed whether all data/samples were fully anonymized before you accessed them and/or whether the IRB or ethics committee waived the requirement for informed consent.

If patients provided informed written consent to have data/samples from their medical records used in research, please include this information.

4. In the ethics statement in the manuscript and in the online submission form, please provide additional information about the patient records/samples used in your retrospective study, including the date range (month and year) during which patients' medical records/samples were accessed.

'I have read the journal's policy and the authors of this manuscript have the following competing interests: According to ICMJE criteria, TPH declares unrestricted research funding for an unrelated project on precision medicine from Roche. YA reports unrelated research grants from the Canadian Institutes of Health Research, the Canadian Dermatology Foundation, AllerGen NCE, Sanofi Canada, Pfizer, Abbvie, and Novartis and honoraria from Janssen, Leo Pharma, Abbvie, Sanofi, Eli Lilly, Pfizer, and Novartis.'

a. Please confirm that this does not alter your adherence to all PLOS ONE policies on sharing data and materials, by including the following statement: "This does not alter our adherence to  PLOS ONE policies on sharing data and materials.” (as detailed online in our guide for authors http://journals.plos.org/plosone/s/competing-interests).  If there are restrictions on sharing of data and/or materials, please state these.

Please note that we cannot proceed with consideration of your article until this information has been declared.

Reviewers' comments:

Reviewer's Responses to Questions

**Comments to the Author**

1. Is the manuscript technically sound, and do the data support the conclusions?

Reviewer #1: Yes

Reviewer #2: Yes

2. Has the statistical analysis been performed appropriately and rigorously? 

Reviewer #1: Yes

Reviewer #2: Yes

3. Have the authors made all data underlying the findings in their manuscript fully available?

Reviewer #1: Yes

Reviewer #2: Yes

4. Is the manuscript presented in an intelligible fashion and written in standard English?

Reviewer #1: Yes

Reviewer #2: Yes

5. Review Comments to the Author

Reviewer #1: Very well written article. It is highlighting a major health concern we have encountered during the current pandemic. Hence I think the topic is very relevant. I think similar studies if undertaken in other countries would show the same decline during the cOVID-19 pandemic. Hence I think its results can be generalized to other countries and populations. Authors have used technically sound data analysis and interpretation.

Reviewer #2: This study presents an interesting observation concerning a drop in skin biopsy for cancer during the COVID pandemic. These results are largely expected based on restrictions on access to heath care during the acute phase of the pandemic. It is not surprising that the drop affected older people more than younger ones, but it is less clearly interpretable why it affected women more than men. How the delay translated into more advanced cancer stage, deaths or complications due to cancer was not explored. This is a major limitation of the study. In addition, no information was provided about what happened in the period of time subsequent to the pandemic more acute phase (namely, after May 2020): was an increase in the number of skin biopsies observed compared to the expected numbers to compensate for the reduction of interventions during the acute phase? what was the profile of patients treated thereafter?

In my opinion, in a highly dynamic situation such as the pandemic, a more comprehensive view on trends and attitudes should be obtained to better understand how healthcare changes affect outcomes

6. PLOS authors have the option to publish the peer review history of their article (what does this mean?). If published, this will include your full peer review and any attached files.

Reviewer #1: **Yes: **Priyanka Bhandari

Reviewer #2: **Yes: **Luigi Naldi

---

## [Author Response · Author response to Decision Letter 0]

24 Feb 2021

Thank-you to the Editor and Reviewers for their evaluation and comments on our article. We have uploaded a file with our responses to the Editor and Reviewers.

---

## [Decision Letter · Decision Letter 1]

1 Mar 2021

Impact of the COVID-19 pandemic on skin cancer diagnosis: a population-based study

PONE-D-20-37376R1

Dear Dr. Hanna

We’re pleased to inform you that your manuscript has been judged scientifically suitable for publication and will be formally accepted for publication once it meets all outstanding technical requirements.

Kind regards,

Mauro Picardo, M.D.

Academic Editor

PLOS ONE

---

## [Editor Report · Acceptance letter]

19 Mar 2021

PONE-D-20-37376R1 

Impact of the COVID-19 pandemic on skin cancer diagnosis: a population-based study 

Dear Dr. Hanna:

I'm pleased to inform you that your manuscript has been deemed suitable for publication in PLOS ONE. Congratulations! Your manuscript is now with our production department. 

Kind regards, 

on behalf of

Dr. Mauro Picardo 

Academic Editor

PLOS ONE